# How to Obtain a High Quality ctDNA in Lymphoma Patients: Preanalytical Tips and Tricks

**DOI:** 10.3390/ph14070617

**Published:** 2021-06-26

**Authors:** Estelle Bourbon, Vincent Alcazer, Estelle Cheli, Sarah Huet, Pierre Sujobert

**Affiliations:** 1Hospices Civils de Lyon, Hôpital Lyon Sud, Service D’hématologie Biologique, 69495 Lyon, France; estelle.bourbon@chu-lyon.fr (E.B.); vincent.alcazer@chu-lyon.fr (V.A.); estelle.cheli@chu-lyon.fr (E.C.); sarah.huet@chu-lyon.fr (S.H.); 2Lymphoma Immunobiology Team, Faculté de Médecine et de Maïeutique Lyon Sud Charles Mérieux, Université Lyon 1, 69921 Oullins, France

**Keywords:** ctDNA, lymphoma, preanalytical

## Abstract

The analysis of circulating tumor DNA (ctDNA) released by tumor cells holds great promise for patients with lymphoma, to refine the diagnostic procedure, clarify the prognosis, monitor the response to treatment, and detect relapses earlier. One of the main challenges of the coming years is to adapt techniques from highly specialized translational teams to routine laboratories as this requires a careful technical and clinical validation, and we have to achieve this as fast as possible to transform a promising biomarker into a routine analysis to have a direct consequence on patient care. Whatever the analytical technology used, the prerequisite is to obtain high yields of ctDNA of optimal quality. In this review, we propose a step-by-step description of the preanalytical process to obtain high-quality ctDNA, emphasizing the technical choices that need to be made and the experimental data that can support these choices.

## 1. Introduction

In their first description of cell-free nucleic acids in blood samples of humans, Mandel and Métais noticed that the level of cell-free DNA (cfDNA) was higher in pregnant women and cancer patients [1]. It took more than 50 years to confirm the clinical value of the analysis of cfDNA to study fetal DNA and to monitor cancer patients, which are still the most widespread applications of cfDNA. Cell-free DNA is thought to be released from cells undergoing apoptosis or necrosis [2,3], mostly leukocytes and endothelial cells [4]. In patients with cancer, especially in the case of high tumor burden or metastatic disease, a large fraction of cfDNA comes from cancer cells; this is named circulating tumor DNA (ctDNA). Among all the nucleic acids released by cancer cells (short fragments, long fragments, mitochondrial DNA, RNA), the short fragments of less than 170 nucleotides (corresponding to the length of DNA in one nucleosome) are the most studied for their use in the clinical setting (for a detailed review, see [5]).

Regarding lymphomas, studies have pointed out the clinical value of ctDNA to refine the diagnosis, to identify predictive biomarkers of response to therapy, to monitor response to treatment, and to detect relapses earlier [6,7,8,9]. One of the biggest challenges in this field is to move from proof-of-concept studies to a real-life setting in order to improve the outcome of lymphoma patients. Careful technical and clinical validation are warranted to adapt ctDNA analysis from highly specialized translational teams to routine laboratories, and we have to achieve this as fast as possible to transform a promising biomarker into a routine analysis to have a direct consequence on patient care. Recently, a study comparing eight up-to-date ctDNA analysis platforms has demonstrated that it remains technically challenging to detect variants below the 0.5% sensitivity threshold [10]. Given this analytical limitation, improving the preanalytical steps to recover high levels of ctDNA of optimal quality is of utmost importance. In this review, we propose a step-by-step description of the preanalytical process to obtain high quality ctDNA, emphasizing the technical choices that need to be made and the experimental data that can support these choices (Figure 1). We have highlighted the technical choices that can be recommended based on our review of the literature when sufficient level of evidences are available.

## 2. The Right Time

Timing of sample collection is an important parameter for ctDNA analysis. The goal of the ctDNA analysis will differ according to the moment of sample collection, from mutation detection before therapy, to molecular response assessment and early detection of relapse or treatment resistance [11].

In Diffuse Large B-Cell Lymphoma (DLBCL), it has been shown that the amount of ctDNA at the time of diagnosis is a surrogate marker of disease burden and harbors prognostic value. Moreover, the decrease of ctDNA concentration after treatment can be used to appreciate the depth of response, either after one (early molecular response) or two (major molecular response) cycles of immunochemotherapy [6]. Another interesting possibility could be to use ctDNA to monitor the clonal composition of cells dying under treatment by performing ctDNA analysis the day after chemotherapy administration. Interesting results came from the longitudinal analysis of ctDNA in lung cancer patients receiving inhibitor of epidermal growth factor receptor (EGFR); in 4/21 patients, Kato et al. detected an increase in ctDNA with unique mutations not identified in the initial biopsy [12]. While these variations were only observed in a very low number of patients and could be associated with daily fluctuations, these observations suggest that ctDNA might capture the mutational signature of subclones particularly sensitive to chemotherapy. Whether the evaluation of response should rely on a few time points or would be better assessed with repeated evaluation after each cycle of immunochemotherapy is an open question for the years to come, keeping in mind the necessity to find the optimal tradeoff from a health economics perspective.

Additional complexity comes from significant intra-individual variations observed from day to day and even hour to hour. For instance, by studying ctDNA concentration variations in 11 lung cancer patients with stable disease and no current therapy, Hojbjerg et al. observed intra-individual day-to-day variations in ctDNA levels ranging from 21% to 53% [13]. Differences were also observed in samples collected at 1 h intervals, with hour-to-hour variations in ctDNA levels ranging from 1% to 47%. In another study, the same group reported slightly different results, with negligible day-to-day variation in cfDNA levels but significant decline according to the time of the day in 10 lung cancer patients and 33 healthy subjects [14]. Nychtemere-associated variations were also described in a study comparing cfDNA level in plasma samples collected four times a day (6 a.m., 12 p.m., 6 p.m., and 12 a.m.) from nine colorectal cancer patients. Up to 50% of variation of the ctDNA levels were noted, with lowest cfDNA levels observed at late timepoints (6 p.m. and 12 a.m.) for early-stage (I–II) cancers, but earlier for stage-IV disease [15].

## 3. The Right Sample

Among the many analytical steps for accurate ctDNA evaluation, the choice of the sample source is of great importance; liquid biopsy encompasses the study of a large variety of body fluids (including peripheral blood, urine, cerebrospinal fluid [CSF], or pleural effusion) that may contain ctDNA released by apoptotic or necrotic tumor cells [16].

### 3.1. Blood Draw

Given that lymphoma is a hematological malignancy arising from lymphocytes circulating in the bloodstream and lymphatic system, blood is likely the best liquid biopsy for this disease.

In the large majority of published studies, the peripheral blood sampled from venipuncture is used for ctDNA analysis in lymphoma patients. However, only few details on the blood draw procedure are generally described, and comparative data of samples acquired from other punction sites, mainly central veins and arteries, are scarce. Förnovick et al. reported in a small cohort of 31 breast cancer patients scheduled for neoadjuvant treatment a significant increase of ctDNA in central blood after breast compression (*p* = 0.01), that was not observed in peripheral blood [17]. In the absence of similar studies in lymphoma patients, blood collected from venous punction remains the reference due to its simplicity, minimal invasiveness, and suitability for routine practice.

### 3.2. Other Fluids

Liquid biopsy is not limited to plasma and recent studies have demonstrated the presence of tumor DNA in other fluids and their application to the diagnosis, screening, and monitoring of cancers [18]. Several non-blood fluids might present advantages over plasma, including better feasibility in routine practice (i.e., urine) or increased sensitivity owing to enriched tumor DNA due to the closer contact with the tumor and the lower dilution with genomic DNA from leucocytes (i.e., CSF).

### 3.3. Urine

Urine is an ideal body fluid for liquid biopsy as it can be collected in a truly non-invasive manner, enabling self-sampling at home. While in the circulation, cfDNA is filtrated from the blood into the primary urine throug the glomerular barrier, which has been proved to be permeable to highly fragmented ctDNA molecules [19]. Two fractions of urinary cfDNA have been described: high molecular weight genomic DNA fragments (>500 bp) originating from urinary tract and endothelial cells, and low molecular weight DNA fragments (50–250 bp) from the circulation [20].

Various studies have reported that genomic alteration of non-urological malignancies such as lung cancer, colorectal cancer, pancreatic cancer or systemic histiocytic disorders, can be identified in urinary cfDNA. When comparing urine to blood samples, other studies demonstrated the good agreement for the detection of EGFR mutation in lung cancer patients in these two body fluids [21,22]. Additionally, Reckamp et al. also reported that a significant decrease in urine EGFR-mutation level correlated with tumor response in patients with lung cancer [21]. Unfortunately, there is no published study of ctDNA in urine in patients with lymphoma.

### 3.4. Cerebrospinal Fluid

The CSF is a very a highly relevant source of ctDNA for primary central nervous system lymphoma (PCNSL). In these cases, CSF ctDNA is particularly appealing to avoid invasive surgical biopsies, to assess the genomic alterations of the tumor, and monitor treatment response. Interestingly, ctDNA is enriched in the CSF as compared to the blood, because the blood–brain barrier limits leukocyte contamination of the CSF and tumor DNA shedding into the peripheral bloodstream. Bobillo et al. demonstrated in a cohort of 19 patients (with PCNLS or systemic lymphoma with cerebral involvement) that the analysis of ctDNA from CSF was more sensitive than ctDNA from blood at diagnosis and allowed for the detection of residual disease better than flow cytometry [23]. Other studies have shown higher sensitivity for mutation detection in CSF than in plasma in brain tumor patients with low systemic metastatic burden [24,25]. Moreover, in a small cohort of patients with brain tumors, De Mattos-Aruda et al. demonstrated a good correlation between tumor DNA load in CSF and response to local or systemic therapy [26].

## 4. The Right Volume

The volume of plasma to be collected will theoretically determine the quantity of cfDNA obtained after extraction. Concentrations of ctDNA in plasma are usually expressed in haploid genome equivalents per mL (hGE/mL) and might be calculated by multiplying the mean variant allele frequency by the input concentration of cfDNA (in pg/mL) and then divided by 3.3 (as one hGE corresponds to 3.3 pg of DNA). However, in the context of cancer, median levels of ctDNA vary substantially across lymphoma subtypes and according to the clinical stage, as well as during the course of the disease, as highlighted below.

At the time of diagnosis, the goal of ctDNA analysis is to define the mutational landscape of the lymphoma, in order to refine the diagnosis, to assess the prognosis or to search for mutations with theranostic value. In this setting, the quantity of plasma that needs to be collected is mostly dependent on the lymphoma subtype; ctDNA concentrations ranging approximately from 20 hGE/mL in follicular lymphoma (FL), 200 hGE/mL in DLBCL to 2000 hGE/mL in primary mediastinal B-cell lymphoma (PMBL), with non-negligible interpatient variations in each entity [6,7,27]. Moreover, within a same lymphoma entity the clinical stage also impacts ctDNA levels as higher levels are reported in stage III-IV DLBCL compared to stage I-II [7,28].

Table 1 represents the number of mutated ctDNA molecules that might be expected according to the shedding ability of the different entities, and variant allele frequency (VAF) of the mutation among the tumor population. The theoretical volume of plasma to be collected will vary considerably whether only clonal events are targeted, or if sub-clonal events with low VAFs are also of interest because they might impact prognosis and/or therapeutic response [29]. As an example, a technique with a sensitivity level of 20 mutated copies will allow the detection of 1% VAF mutations starting from 1 mL of plasma in a tumor shedding 2000 hGE/mL, but will be able to detect only 10% VAF mutations starting from 10 mL of plasma in a tumor shedding 20 hGE/mL. Of note, the detected VAFs will be impacted by dilution in other circulating DNA, such as non-tumor cfDNA or contaminating genomic DNA from leukocytes lysis (see below).

In the context of monitoring residual disease, the volume of plasma (and thus ctDNA quantity) represents a potential limitation to the overall sensitivity of the analysis. Even with a very sensitive technique able to detect one target mutation among 10^5^ unmutated reads, the sensitivity of the analysis would not exceed 10^−4^ if only 10^4^ hGE have been sampled (corresponding to 5 mL of plasma in the “ctDNA-rich” entity PMBL). As elegantly demonstrated by Zviran et al., increasing the number of mutational targets for minimal residual disease (MRD) monitoring increases the sensitivity of the analysis, even if the technical sensitivity is low. Hence, even with a moderate plasma sample volume, as low as 1 mL, very deep sensitivity is achievable with whole genome sequencing [30]. Taken together, the determination of the volume of plasma to be sampled is a trade-off between what is expected from the test and what analytical technique is planed. It seems reasonable to limit the quantity of blood sampling to avoid harmful consequences on the patient [31]; in routine practice we therefore recommend to draw 20 mL of blood (thus approximately 10–12 mL of plasma), although the analytical sensitivity will be impacted in some cases with low ctDNA levels (e.g., follicular lymphoma) as presented in Table 1.

## 5. The Right Tube

One of the key factors to obtain high-quality ctDNA is to avoid the release of DNA from leukocytes lysis after collection of the blood sample. The level of genomic DNA released from leukocytes might be as much as 10-fold higher than the cfDNA, thus diluting the useful information possibly under the detection threshold for low allelic frequency variants. Another concern is the release of DNAse activity, which might alter the quality of cfDNA. Furthermore, excessive lysis of white blood cells can substantially increase the cost of sequencing, and increase the rate of false-positive results due to clonal hematopoiesis of indeterminate potential [4].

Several studies have reported that the addition of stabilizers in the collection tubes might overcome this important barrier. Many suppliers propose commercial solutions such as BCT (Streck, La Vista, NE, USA), PAXgene (Qiagen, Hilden, Germany), Cell-Free DNA Collection Tube (Roche, Basel, Switzerland), or CellSave preservative tube (Janssen Diagnostics, Raritan, NJ, USA). All published studies comparing these to EDTA tubes have consistently demonstrated their superiority to prevent white blood cell lysis (Table 2), with no major difference between stabilizer tubes. Notably, EDTA tubes might be used if the plasma is separated within a short period of time (<4–6 h at room temperature, or <48 h at +4 °C) [32,33]. It is, however, of note that all improved-performance stabilizer tubes are 50 to 100 times more costly than EDTA tubes, which might limit their use in low-resource settings.

As far as we know, there is no published study reporting rigorous evaluation of the different stabilizer tubes for the analysis of ctDNA in patients with lymphoma. Accordingly, it seems reasonable to use any of the commercial stabilizer tubes, and to avoid EDTA tubes except if prompt plasma separation is possible.

## 6. The Right Shipping

Transportation through pneumatic systems can reduce the time between blood sampling and processing, but acceleration and decelerations might also alter the integrity of white blood cells and dilute the ctDNA. The effects of pneumatic tube transportation on cfDNA were initially evaluated in healthy volunteers, using EDTA collecting tubes, and did not detect any significant difference in the amount of cfDNA as assessed by quantitative PCR [34]. Another study on healthy volunteers reported that shaking of the tubes increased the amount of cfDNA if the blood was sampled in EDTA tubes, but not in BCT tubes containing a stabilizer agent [35]. In cancer patients, a comparative study of 25 patients with esophageal cancer assessed by ddPCR have convincingly shown that pneumatics increases DNA release from leucocytes in EDTA sampling, but not in tubes with stabilizers [36]. Altogether, we can conclude from these studies that pneumatic transportation systems can be used safely for ctDNA analysis as long as stabilizers containing tubes are used, but this might increase ctDNA dilution in EDTA tubes.

## 7. The Right Blood Separation Protocol

Numerous different centrifugation protocols have been proposed for plasma processing to provide cell-free DNA, and these are reviewed in [37]. These protocols vary greatly regarding the number of centrifugation steps, their speed (i.e., centrifugal force), and duration. A protocol with double centrifugation is largely consensual [32,38], associating a first round at low speed to remove leucocytes while minimizing cell lysis, and a second high-speed centrifugation step to pellet any residual cells or cellular debris. Samples subjected to two rounds of centrifugation contain up to 60-fold less DNA than plasma samples that are centrifuged only once prior to DNA extraction [39]. Current recommendations advise a first centrifugation at 820–1600× *g* for 20 min and a second centrifugation performed either at 14,000–16,000× *g* for 10 min or 3000–6000× *g* for 20 min [32,38].

If necessary, plasma can be frozen at −80 °C before DNA extraction without excessive alteration of the yield and quality of cfDNA [32,38]. However, as plasma freeze–thaw events induce DNA fragmentation and alter cfDNA quality, it is preferable to aliquot plasma into single-use fractions before DNA extraction if it is not performed immediately, but there is no consensus on the maximum acceptable plasma storage duration [38].

## 8. The Right cfDNA Extraction Method

After plasma isolation, the next critical step is cfDNA extraction that must also be optimized for the recovery of short fragments. The main criteria for the evaluation of the extraction systems is the amount of cfDNA obtained in order to maximize the sensitivity of the entire procedure. Although column-based solutions are more popular, magnetic beads might be interesting to increase the input plasma volume. Another important consideration is the possibility of automation and multiplexing of the extraction procedure. A lot of commercial solutions are available on the market, and some of them have been compared in benchmark studies in different clinical settings with different primary endpoints [40,41,42,43,44]. To our knowledge, there is no such comparative study for the extraction of ctDNA in lymphoma patients, but there is no obvious reason to assume that lymphoma ctDNA extraction should be different from other cancer ctDNA. For a detailed review about this topic, the reader is referred to a recently published review [45].

## 9. The Right Way to Assess ctDNA Quality

Whatever the analytical technique used, it is important to quantify the total amount of plasmatic DNA (including DNA from leukocyte lysis) because analytical techniques measure a ratio of mutated to wild-type alleles, which might be artificially lowered if the ctDNA is diluted. A precise quantification of the total plasmatic DNA enables to transform this ratio in an absolute quantity of mutated variant. Hence, the absolute number of mutated ctDNA will be the same with a variant allele frequency of 10% in 3000 hGE (300 hGE) and if the same quantity of ctDNA is diluted in two-fold more normal DNA (VAF 5% in 6000 hGE = 300 hGE). Of note, the small amounts of cfDNA constrains the choice of the quantification technique. The absolute nucleic acid quantification might rely spectrophotometry or fluorimetry, or on quantitative PCR of a housekeeping gene such as albumin. Given the small size of ctDNA fragments, the PCR primers have to be designed to target small-size amplicons. Further refinements have been proposed with the analysis of different PCRs with different amplicon sizes, based on the idea that cfDNA will be mainly amplified with short amplicon PCR (41 bp), whereas long amplicons will also be amplified in DNA released from cell lysis. Using a commercial solution (hgDNA quantification & QC kit, Kapa/Roche), Nikolaev et al. have demonstrated that the PCR of small amplicons is useful for cfDNA quantification, and the ratio of long/small amplicons is a good indicator of contamination by cell lysis [33]. Another possibility is to use an electrophoresis-based analyzer, but these devices usually require a minimum amount of DNA that might limit its applicability for ctDNA.

## 10. The Right Way to Conserve ctDNA

After extraction, ctDNA can be store at −20 °C but studies have suggested an alteration of ctDNA over time; the estimated yearly decay is 30% [46,47]. These considerations are particularly important in case of retrospective analysis of samples from clinical trials, or for the use of samples as internal quality control.

## 11. Conclusions

The analysis of ctDNA holds great promise for lymphoma patients, both at the time of diagnosis as well as during follow-up. However, transforming this hope into a clinically significant improvement will require a huge effort of standardization and quality assessment. Here we have explicitly described the main technical choices regarding preanalytical steps that might impact the quality of ctDNA analysis. Of note, most of the data discussed here come from studies in patients with solid cancer. Even if it is highly probable that these results might be relevant also for lymphoma patients, dedicated studies would be important to confirm at least the most critical steps. In the absence of these studies, we encourage the use of internal quality controls to ensure detectability of technical drift. We also recommend detailed reporting of the preanalytical methods in articles dealing with ctDNA in lymphoma patients.

Once a high-quality ctDNA is obtained, the analytical steps are also essential to ensure quantitative, reproducible, sensitive, and specific results. The most critical questions are the panel design, to provide an optimal informativity/panel size ratio, the technical choices regarding library preparation (amplicon vs. capture, use of unique molecular identifiers), and optimization of the bioinformatics pipeline. Then, the reporting of ctDNA analysis should be standardized following international recommendations in order to facilitate their use for clinicians.

ctDNA analysis is also challenging for quality assessment programs and accreditation. Basic questions such as the nature of the matrix for external quality assessment remain unsolved. An ideal matrix would be blood samples from patients, but given the low levels of ctDNA, it would require too large volume to send to a sufficient number of laboratories. The use of plasma samples with spike-in DNA has been reported, as well as cell-line derived DNA or sequencing data, all of them testing the analytical procedure with different caveats [10,48]. Alternatively, the development of commercial diagnostic solutions rigorously evaluated in international consortiums might be an interesting solution to harmonize this analysis worldwide. Initiatives such as the precommercial procurement OncNGS (http://oncngs.eu, accessed on 11 May 2021) developed by the European Community are probably going to respond to this unmet need.

## Figures and Tables

**Figure 1 pharmaceuticals-14-00617-f001:**
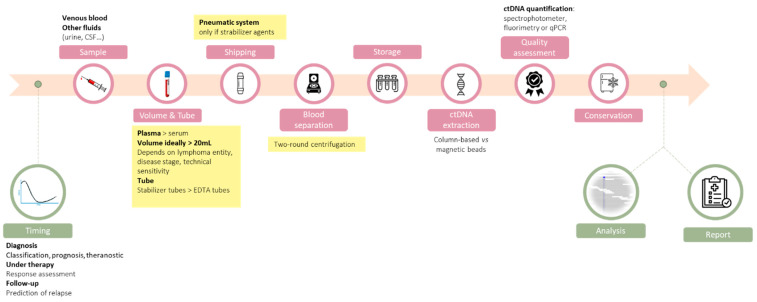
Overview of the ctDNA analysis workflow, with an emphasis on the preanalytical steps that are detailed in this review. When sufficient levels of evidence exist, our recommendations are highlighted in yellow.

**Table 1 pharmaceuticals-14-00617-t001:** Number of mutated ctDNA molecules that might be expected according to the volume of plasma collected, and the tumor and variant allele fraction (VAF) of the mutation. Three situations are envisaged: top: ctDNA = 2000 hGE/mL (e.g., 4000 hGE/mL of total cfDNA with 50% being tumor-derived); middle: ctDNA = 200 hGE/mL (e.g., 400 hGE/mL of total cfDNA with 50% being tumor-derived); bottom: ctDNA = 20 hGE/mL (e.g., 40 hGE/mL of total cfDNA with 50% being tumor-derived). Color key according to the number of ctDNA copies harboring the variant (red 0–10; orange 11–100; green > 100).

**ctDNA Concentration of 2000 hGE/mL** **(e.g., PMBL), Corresponding to 6.6 ng/mL**	**Volume of Plasma (mL)**	**Haploid Genome Equivalents**	**ctDNA Quantity (ng)**	**Number of Mutated ctDNA Copies (hGE) at Different VAFs**				
				**VAF 50%**	**VAF 10%**	**VAF 1%**	**VAF 0.1%**	**VAF 0.01%**
	50	100,000	330	50,000	10,000	1000	100	10
	20	40,000	132	20,000	4000	400	40	4
	10	20,000	66	10,000	2000	200	20	2
	5	10,000	33	50,000	1000	100	10	1
	1	2000	6.6	10,000	200	20	2	0.2
**ctDNA Concentration of 200 hGE/mL** **(e.g., DLBCL), Corresponding to 0.66 ng/mL**	**Volume of Plasma (mL)**	**Haploid Genome Equivalents**	**ctDNA Quantity (ng)**	**Number of Mutated ctDNA Copies (hGE) at Different VAFs**				
				**VAF 50%**	**VAF 10%**	**VAF 1%**	**VAF 0.1%**	**VAF 0.01%**
	50	10,000	33	5000	1000	100	10	1
	20	4000	13.2	2000	400	40	4	0.4
	10	2000	6.6	1000	200	20	2	0.2
	5	1000	3.3	5000	100	10	1	0.1
	1	200	0.66	100	20	2	0.2	0.02
**ctDNA Concentration of 20 hGE/mL** **(e.g., FL), Corresponding to 0.066 ng/mL**	**Volume of Plasma (mL)**	**Haploid Genome Equivalents**	**ctDNA Quantity (ng)**	**Number of Mutated ctDNA Copies (hGE) at Different VAFs**				
				**VAF 50%**	**VAF 10%**	**VAF 1%**	**VAF 0.1%**	**VAF 0.01%**
	50	1000	3.3	500	100	10	1	0.1
	20	400	1.32	200	40	4	0.4	0.04
	10	200	0.66	100	20	2	0.2	0.02
	5	100	0.33	50	10	1	0.1	0.01
	1	20	0.066	10	2	0.2	0.02	0.002

**Table 2 pharmaceuticals-14-00617-t002:** Summary of the main comparative studies of blood collection tubes.

	Study Design	Origin of Samples	Delay	Parameter	Main Conclusion
**Parackal, 2019**	Streck vs. Roche	healthy donors	up to 14 days	DNA yield and quality	no difference
**Alidousty, 2017**	Streck vs. Roche vs. Qiagen	healthy donors	up to 7 days	PCR	no difference
**Zhao, 2018**	EDTA vs. Streck vs. Roche	healthy donors	up to 14 days	PCR and NGS	Streck or Roche better than EDTA. Roche better than Streck at d14
**Medina Diaz, 2016**	EDTA vs. Streck	colorectal cancer	up to 5 days	PCR	Streck better than EDTA, precision about the shipping temperature
**Kang, 2016**	EDTA vs. Streck vs. Cellsave	metastatic breast cancer	2 days	ddPCR	Streck and CellSave better than EDTA
**Sacher, 2016**	EDTA vs. Streck	lung cancer, prospective trial	overnight	DNA yield and ddPCR	Streck at room temperature = EDTA on ice
**Norton, 2013**	EDTA vs. Streck	healthy donors	up to 14 days	ddPCR	Streck > EDTA

## Data Availability

Data sharing not applicable.

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
