# Peer review of "How to Obtain a High Quality ctDNA in Lymphoma Patients: Preanalytical Tips and Tricks"

_pharmaceuticals, 2021, doi:10.3390/ph14070617_

Round 1

Reviewer 1 Report

This review could be better written and falls short of being really useful for most readers. The authors are trying to balance a line between a protocol and a review with little depth and I am not convinced that works well. In particular, in some instances there are no references. In other cases information is only vaguely touched on (such as line 33 “very convincing studies” or line 42 “multiple platforms”). My feeling is, either the authors should go into more detail over and above these statements and the correlating references or avoid such comments altogether.

Even in areas where more information is given, that is usually not linked to any critical review how good the underlying data are, analysis of the literature mostly remains in a descriptive state. For example for sentence 61-64 reference 2 is provided as justification for proposing analysis of ctDNA changes associated with response. However, there is no mention what the data of that study are and how convincing they are. There, dynamics of ctDNA from 217 patients was analysed so it can be considered a good study to lend evidence to the idea of ctDNA as response monitor. On the other hand, reference 7 does not really justify the statement (line 64-66) “…could be the analysis of ctDNA the day after chemotherapy administration, in order to monitor the clonal composition of the cells dying under treatment.” Also the data of that study are less convincing especially the ctDNA detection increase immediately after treatment is only seen in few patients and on inspection could possibly fall into variation between samples and ctDNA detectability (as the authors do discuss in fact later: day-to-day variability). Moreover, the issue of a detectable ctDNA increase (as monitor) is debated in the research field and a critical review should consider this.

 Major issues to address:

  1. Most of the statements in the first section are not referenced. Add references.
  2. Line 33: While it is encouraging that the authors find ctDNA studies in lymphoma convincing, in a [critical] review it is more relevant to point out the actual findings and evaluate how/why they are convincing or not.
  3. Line 33-34: “demonstrated the interest of ctDNA to…” what do you mean? Re-write.
  4. Line 60: acronym DLBCL has not been introduced
  5. Line 74: clarify what is meant with “served up to 53%/19% day-to-day and 27%/56%”
  6. Line 75: who is “they”
  7. Section line 97-101: in the context it is not quite clear why the authors discuss CTCs here, given they focus on ctDNA. There is a principal difference between CTCs and ctDNA that could well be discussed here, but in the current manuscript it is simply thrown in without context: Clearly CTCs are cells and by size and plasticity [at least for solid cancers] not designed to be in circulation. Thus, they get filtered out in small capillaries (ie CTC numbers decrease fast) and it makes sense that collection of blood close to the source will deliver more CTCs. ctDNA on the other hand, even if harvested by including exosomal DNA, would not be expected to be filtered out, thus, assuming a reasonable ctDNA stability peripheral blood should not have a significantly different ctDNA content. Either rewrite this section accordingly or remove mention of CTC data.
  8. Section line 106-143: while there is potential utility of “other” liquid biopsies in some cancers, given that lymphoma is a blood cancer, blood is likely the best liquid biopsy for this disease. As such discussion of other liquid biopsies is of little relevance here. If the authors want to include this section they should at least discuss how these various biopsies may be useful in lymphoma [or not].
  9. Line 151-153: sentence not referenced
  10. Figure 2 is not a figure, a set of tables are presented

Author Response

REVIEWER 1

This review could be better written and falls short of being really useful for most readers. The authors are trying to balance a line between a protocol and a review with little depth and I am not convinced that works well.

We deeply regret that Reviewer 1 did not found enough interest in this review. Indeed, it has been written in accordance with the editorial request. We have tried to improve the manuscript following his/her comments.

In particular, in some instances there are no references. In other cases information is only vaguely touched on (such as line 33 “very convincing studies” or line 42 “multiple platforms”). My feeling is, either the authors should go into more detail over and above these statements and the correlating references or avoid such comments altogether.

We agree with the reviewer that the content of this review is a little bit superficial regarding the clinical value of ctDNA in lymphomas. However, we argue that this review is a chapter of a whole volume of Pharmaceuticals dedicated to ctDNA in lymphoma. Accordingly, we believe that the introduction has to recall the global context of ctDNA analysis in lymphomas, and the reader is referred to the important papers in the field which will be commented with more details in other reviews of the same volume.

Even in areas where more information is given, that is usually not linked to any critical review how good the underlying data are, analysis of the literature mostly remains in a descriptive state. For example for sentence 61-64 reference 2 is provided as justification for proposing analysis of ctDNA changes associated with response. However, there is no mention what the data of that study are and how convincing they are. There, dynamics of ctDNA from 217 patients was analysed so it can be considered a good study to lend evidence to the idea of ctDNA as response monitor.

We fully agree with the reviewer that this is a major paper in the field of ctDNA analysis of DLBCL, but here again this paper will be discussed in another review of the same volume. Here we wanted to focus on the preanalytical steps, not on the clinical value analysis (even if we had to mention it of course).

On the other hand, reference 7 does not really justify the statement (line 64-66) “…could be the analysis of ctDNA the day after chemotherapy administration, in order to monitor the clonal composition of the cells dying under treatment.” Also the data of that study are less convincing especially the ctDNA detection increase immediately after treatment is only seen in few patients and on inspection could possibly fall into variation between samples and ctDNA detectability (as the authors do discuss in fact later: day-to-day variability). Moreover, the issue of a detectable ctDNA increase (as monitor) is debated in the research field and a critical review should consider this.

We fully agree with the reviewer that this sentence has to be nuanced. The manuscript has been modified accordingly with the following paragraph:

"Another interesting possibility could be to use ctDNA to monitor the clonal composition of cells dying under treatment by performing ctDNA analysis the day after chemotherapy administration. Interesting results came from the longitudinal analysis of ctDNA in lung cancer patients receiving inhibitor of epidermal growth factor receptor.  In 4 out of 21 patients, Kato et al detected an increase in ctDNA with unique mutations not identified in the initial biopsy (Kato et al., 2016). While these variations were only observed in a very low number of patients and could be associated with daily fluctuations, these observations suggest that ctDNA might capture the mutational signature of subclones especially sensitive to chemotherapy"

Major issues to address:

Most of the statements in the first section are not referenced. Add references.

We added four references in this section.

Line 33: While it is encouraging that the authors find ctDNA studies in lymphoma convincing, in a [critical] review it is more relevant to point out the actual findings and evaluate how/why they are convincing or not.

We of course agree with the reviewer that a critical review of the value of ctDNA in lymphomas is required. However, as already pointed previously, this review is part of a special issue of Pharmaceuticals dedicated to the analysis of ctDNA in lymphomas, so we do not believe that an extensive critical discussion of these papers is warranted in this review about the preanalytical aspects

Line 33-34: “demonstrated the interest of ctDNA to…” what do you mean? Re-write.

We have rephrased this sentence as follow "Regarding lymphomas, very convincing studies have pointed out the clinical value of ctDNA to refine the diagnosis, to identify predictive biomarkers of response to therapy, to monitor response to treatment, and to detect relapses earlier”

Line 60: acronym DLBCL has not been introduced

Thank you, modification has been done.

Line 74: clarify what is meant with “served up to 53%/19% day-to-day and 27%/56%”

Line 75: who is “they”

The whole paragraph has been clarified as follow:

Additional complexity comes from significant intra-individual variations observed from day-to-day and even hour-to-hour. For instance, by studying ctDNA concentration variations in 11 lung cancer patients with stable disease and no current therapy, Hojbjerg et al. observed intra-individual day-to-day variations in ctDNA levels ranging from 21% to 53% [13]. Differences were also observed in samples collected at 1-hour intervals, with hour-to-hour variations in ctDNA levels ranging from 1% to 47%. In another study, the same group reported slightly different results, with negligible day-to-day variation in cfDNA levels but significant decline according to the time of the day in 10 lung cancer patients and 33 healthy subjects

Section line 97-101: in the context it is not quite clear why the authors discuss CTCs here, given they focus on ctDNA. There is a principal difference between CTCs and ctDNA that could well be discussed here, but in the current manuscript it is simply thrown in without context: Clearly CTCs are cells and by size and plasticity [at least for solid cancers] not designed to be in circulation. Thus, they get filtered out in small capillaries (ie CTC numbers decrease fast) and it makes sense that collection of blood close to the source will deliver more CTCs. ctDNA on the other hand, even if harvested by including exosomal DNA, would not be expected to be filtered out, thus, assuming a reasonable ctDNA stability peripheral blood should not have a significantly different ctDNA content. Either rewrite this section accordingly or remove mention of CTC data.

We thank the reviewer for this important comment highlighting that CTC and ctDNA are highly different, and that the reference to CTC is not adequate here. Accordingly, we have removed this part of the review. Instead we have added a sentence about a paper dealing with ctDNA differences between central and peripheral veins: " Förnovick et al. reported in a small cohort of 31 breast cancer patients scheduled for neoadjuvant treatment a significant increase of ctDNA after breast compression in central blood (p = 0.01), that was not observed in peripheral testing (Förnvik et al., 2019)"

Section line 106-143: while there is potential utility of “other” liquid biopsies in some cancers, given that lymphoma is a blood cancer, blood is likely the best liquid biopsy for this disease. As such discussion of other liquid biopsies is of little relevance here. If the authors want to include this section they should at least discuss how these various biopsies may be useful in lymphoma [or not].

This is an important point to discuss. Even if they originate from blood cells, lymphomas can disseminate in virtually all organs and tissues. Accordingly, it seems relevant to think about the role of ctDNA analysis of other samples than blood. Indeed, we choose to discuss urine and cerebrospinal fluid (CSF) as these two non-blood fluids might be of clinical interest in lymphoma as discussed in this section. ctDNA analysis in urine presents the advantage of lower invasiveness, and analysis of CSF is more sensitive as compared to blood in the case of primary central nervous system lymphomas, which is a highly relevant clinical problem.

Line 151-153: sentence not referenced

This is an introductory sentence to what is developed in the following paragraphs. The sentence has been modified:

“However, in the context of cancer, median levels of ctDNA vary substantially across lymphoma subtypes and according to the clinical stage, as well as during the course of the disease, as highlighted below”.

Figure 2 is not a figure, a set of tables are presented

The reviewer is right, and we have modified the manuscript accordingly by transforming figure 2 in table 1

Reviewer 2 Report

This review is structured in a very useful way and I liked the headers – the good tube, the good volume etc Unfortunately in many instances, no recommendation is provided, and I think this should be the goal of the study – although we cannot always be certain of the best approach– the authors should be able the synthesise the available information to provide the ready with their recommendations.

Some improvements to grammar are also required, and I suggest a thorough review.

Introduction: There is sufficient uncertainty regarding the source of cfDNA that this statement should be updated. I would suggest - Cell free DNA is thought to be derived from cells undergoing

Introduction. Sentence ‘ Adapting techniques…’ sentence needs improving ...requires careful technical

Introduction. Sentence ‘The analytical difficulties…. This sentence needs revision

Page 2: Can this sentence be clarified - do the authors mean that 53% daily variation in ctDNA and only 19% in cfDNA . Some discussion on why ctDNA would vary much more than cfDNA is worthwhile?

Page 2 4th sentence from end of page: what does 'day time' mean time of day?

Page 4 – The good Volume, last paragraph: The numbers in this sentence are not clear are these meant to read as 10 to the power of 4 and 5??

No recommendation for blood volume is provided.

Figure 2 Should the headers of these tables be cfDNA = 2000hGE/ml and then if the ctDNA VAF is 50% 1000 ctDNA copies available

The price difference between EDTA and stabilised tubes needs to be mentioned

There is no reason to assume that lymphoma ctDNA extraction should be different from other cancer ctDNA = and thus some summary on preferred extraction kit would be valuable

Author Response

REVIEWER 2
This review is structured in a very useful way and I liked the headers – the good tube, the good volume etc. Unfortunately in many instances, no recommendation is provided, and I think this should be the goal of the study – although we cannot always be certain of the best approach– the authors should be able the synthesise the available information to provide the ready with their recommendations.

We thank the reviewer for his/her positive evaluation of this review. We agree that ideally we should be able to offer recommendations, but as underlined by Reviewer 2, the best technical solution is not always clear... However, we have indicated in Figure 1 what our preferred choice is after this review of the literature, when it was possible to make this choice based on objective criteria. A sentence about this point is included in the main text.

Some improvements to grammar are also required, and I suggest a thorough review.

The paper has been fully reviewed by a native English speaker.

Introduction: There is sufficient uncertainty regarding the source of cfDNA that this statement should be updated. I would suggest - Cell free DNA is thought to be derived from cells undergoing

We thank the reviewer for this important point; we have updated the text accordingly.

Introduction. Sentence ‘ Adapting techniques…’ sentence needs improving ...requires careful technical

We have changed the syntax of this sentence: "Careful technical and clinical validation are warranted to adapt ctDNA analysis from highly specialized translational teams to routine laboratories "

Introduction. Sentence ‘The analytical difficulties…. This sentence needs revision

This sentence has been modified for " Recently, a study comparing eight up-to-date ctDNA analysis platforms has demonstrated that it remains technically challenging to detect variants below the 0.5% sensitivity threshold "

Page 2: Can this sentence be clarified - do the authors mean that 53% daily variation in ctDNA and only 19% in cfDNA . Some discussion on why ctDNA would vary much more than cfDNA is worthwhile?

The sentence has been clarified. To avoid misleading, cfDNA variations described in this paper have been removed as the differences compared to ctDNA were not significant and are not relevant in this context. The aim of this paragraph is to underline the day-to-day or hour-to-hour variation in either one of the two marker (ctDNA or cfDNA), and weither ctDNA vary much more than cfDNA is a complex question for which we have no clear data to answer.

Page 2 4th sentence from end of page: what does 'day time' mean time of day?

The sentence has been clarified: "In another study, the same group reported slightly different results, with negligible day-to-day variation in cfDNA levels but significant decline according to the time of the day in 10 lung cancer patients and 33 healthy subjects"

Page 4 – The good Volume, last paragraph: The numbers in this sentence are not clear are these meant to read as 10 to the power of 4 and 5??

The reviewer is right, this is a typo error that is now corrected.

No recommendation for blood volume is provided.

We have now added a sentence in the text (ideally > 20mL blood) and in figure 1.

"Therefore, in routine practice we recommend to sample a volume of 20 mL of blood (thus approximately 10-12 mLof plasma), although the analytical sensitivity will be impacted in in some cases with low ctDNA levels (eg, follicular lymphoma) as demonstrated in Table 1.

Figure 2 Should the headers of these tables be cfDNA = 2000hGE/ml and then if the ctDNA VAF is 50% 1000 ctDNA copies available

This is an important question. As the proportion of ctDNA among total cfDNA is highly variable, we have decided for the reader’s understanding to express these mathematical considerations only as ctDNA concentrations. The whole explanation in “the good volume” section is built on ctDNA quantities/concentrations in the different lymphoma entities, so we did not want to introduce any additional confusing factor. However, given the reviewer's comment, we have decided to precise as follow in the legend of the figure (which is now a table according to reviewer 1 suggestion):

top – ctDNA= 2000 hGE/mL (eg, 4000 hGE/mL of total cfDNA with 50% being tumor-derived)

middle – ctDNA= 200 hGE/mL (eg, 400 hGE/mL of total cfDNA with 50% being tumor-derived)

bottom– ctDNA= 20 hGE/mL (eg, 40 hGE/mL of total cfDNA with 50% being tumor-derived)

The price difference between EDTA and stabilised tubes needs to be mentioned

The reviewer is right, and we have added the following sentence:

"Of note, all the improved performances stabilizers tubes are 50 to 100 times more costly than EDTA tubes, which might limit their use in low-ressource settings."

There is no reason to assume that lymphoma ctDNA extraction should be different from other cancer ctDNA = and thus some summary on preferred extraction kit would be valuable

We have precised the point raised by the reviewer that there is no reason to assume a difference according to cancer type. In this section, we have explicited the pros and cons of the two main mehods, and referred to a recent review. We believe that the choice of the extraction kit should be made by the labs given these elements, and we prefer not to propose one against another.

Round 2

Reviewer 1 Report

The authors have addressed some comments, but argued unconvincingly to not address others:

Authors address of point

  1. “in some instances there are no references. In other cases information is only vaguely touched on (such as line 33 “very convincing studies” or line 42 “multiple platforms”). My feeling is, either the authors should go into more detail over and above these statements and the correlating references or avoid such comments altogether.”

While I see the authors point that their review may be published in a special issue revealing more background, that does not justify not referencing statements. If they use language such as: “very convincing studies” or “multiple platforms”, then referencing is appropriate and part of good scientific presentation. The request for inclusion of references is still active and required for acceptance.

  1. “changes associated with response. However, there is no mention what the data of that study are and how convincing they are. There, dynamics of ctDNA from 217 patients was analysed so it can be considered a good study to lend evidence to the idea of ctDNA as response monitor.”

The authors appear somewhat naïve, if they feel that given another review in the special issue is already discussing certain issues, that are also relevant to their review, justifies not appropriately reviewing them in theirs. Do they seriously believe that every reader actually reads both reviews? The issue still needs to be addressed appropriately.

  1. “While it is encouraging that the authors find ctDNA studies in lymphoma convincing, in a [critical] review it is more relevant to point out the actual findings and evaluate how/why they are convincing or not.”

The same: this comment should be addressed assuming their review as a stand alone manuscript. The issue still needs to be addressed appropriately.

  1. Authors reply: “This is an important point to discuss. Even if they originate from blood cells, lymphomas can disseminate in virtually all organs and tissues. Accordingly, it seems relevant to think about the role of ctDNA analysis of other samples than blood. Indeed, we choose to discuss urine and cerebrospinal fluid (CSF) as these two non-blood fluids might be of clinical interest in lymphoma as discussed in this section. ctDNA analysis in urine presents the advantage of lower invasiveness, and analysis of CSF is more sensitive as compared to blood in the case of primary central nervous system lymphomas, which is a highly relevant clinical problem.”

What is the associated change in the manuscript here? If they state that this is an important issue to discuss, this discussion clearly should go into the review to justify discussion of other liquid biopsies.

  1. Also I forgot to mention before that the title should be changed: “get a good ctDNA in lymphoma patients” should be changed to a more appropriate language.

Author Response

The authors have addressed some comments, but argued unconvincingly to not address others:

Authors address of point

  1. “in some instances there are no references. In other cases information is only vaguely touched on (such as line 33 “very convincing studies” or line 42 “multiple platforms”). My feeling is, either the authors should go into more detail over and above these statements and the correlating references or avoid such comments altogether.”

While I see the authors point that their review may be published in a special issue revealing more background, that does not justify not referencing statements. If they use language such as: “very convincing studies” or “multiple platforms”, then referencing is appropriate and part of good scientific presentation. The request for inclusion of references is still active and required for acceptance.

If we understand well the reviewer’s comment, he/she advise us either to describe the referred studies with more details, or to avoid the use of words that might be interpreted as an inappropriate judgement. In this introduction, we do not want to go into details, but just present an overview of the whole field, because our review is focused on technical aspects of ctDNA analysis.

Accordingly, we have removed the term “very convincing”. The term “multiple platforms” has already been removed for “Recently, a study comparing eight up-to-date ctDNA analysis platforms has demonstrated that it remains technically challenging to detect variants below the 0.5% sensitivity threshold”, and we do not understand what is the problem with this sentence.

Finally, these sentences are supported by respectively 3 and 1 references, and we have already added other references in the introduction paragraph to answer Reviewer 1 request.

2. “changes associated with response. However, there is no mention what the data of that study are and how convincing they are. There, dynamics of ctDNA from 217 patients was analysed so it can be considered a good study to lend evidence to the idea of ctDNA as response monitor.”

The authors appear somewhat naïve, if they feel that given another review in the special issue is already discussing certain issues, that are also relevant to their review, justifies not appropriately reviewing them in theirs. Do they seriously believe that every reader actually reads both reviews? The issue still needs to be addressed appropriately.

While we agree with the reviewer that most of the readers will not read all the reviews of this special issue, we do not share his/her point of view regarding the interest of a more detailed description of this paper (and other, see point 1). Otherwise, we could also argue that we should describe more precisely the mechanisms of ctDNA release, the characteristics of ctDNA and other circulating nucleic acids, and so on. In order to keep this introduction paragraph short enough, we prefer to refer the reader to other reviews on these topics.

“While it is encouraging that the authors find ctDNA studies in lymphoma convincing, in a [critical] review it is more relevant to point out the actual findings and evaluate how/why they are convincing or not.” The same: this comment should be addressed assuming their review as a stand alone manuscript. The issue still needs to be addressed appropriately.

We have removed the term “very convincing”

4. Authors reply: “This is an important point to discuss. Even if they originate from blood cells, lymphomas can disseminate in virtually all organs and tissues. Accordingly, it seems relevant to think about the role of ctDNA analysis of other samples than blood. Indeed, we choose to discuss urine and cerebrospinal fluid (CSF) as these two non-blood fluids might be of clinical interest in lymphoma as discussed in this section. ctDNA analysis in urine presents the advantage of lower invasiveness, and analysis of CSF is more sensitive as compared to blood in the case of primary central nervous system lymphomas, which is a highly relevant clinical problem.”

What is the associated change in the manuscript here? If they state that this is an important issue to discuss, this discussion clearly should go into the review to justify discussion of other liquid biopsies.

We have to apologize to the reviewer, because the changes in the manuscript were not enough clearly highlighted (line 121-124) : “Several non-blood fluids might present advantages over plasma, including better feasibility in routine practice (i.e. urine) or increased sensitivity owing to enriched tumor DNA due to the closer contact with the tumor and the lower dilution with genomic DNA from leucocytes (i.e. CSF).”

5. Also I forgot to mention before that the title should be changed: “get a good ctDNA in lymphoma patients” should be changed to a more appropriate language.

We have changed the title accordingly for “How to obtain a high quality ctDNA in lymphoma patients: preanalytical tips and tricks.”
